# Photocatalytic Activity of Monosized AuZnO Composite Nanoparticles

**Chenguang Ma, Xianhong Wang, Shixia Zhan, Xuemei Li, Xiao Liu, Yun Chai \*, Ruimin Xing \* and Hongling Liu \***

Key Lab of Polyoxometalate Chemistry of Henan Province, Institute of Molecular and Crystal Engineering, School of Chemistry and Chemical Engineering, Henan University, Kaifeng 475001, China; 104753160773@vip.henu.edu.cn (C.M.); 104753120887@vip.henu.edu.cn (X.W.); 104753170761@vip.henu.edu.cn (S.Z.); 104753130832@vip.henu.edu.cn (X.L.); 104753130823@vip.henu.edu.cn (X.L.)

\* Correspondence: chaiyun@henu.edu.cn (Y.C.); xingenjoy@163.com (R.X.); hlliu@henu.edu.cn (H.L.)

**Abstract:** Photocatalytic activity of monosized AuZnO composite nanoparticles with different compositions were synthesized by the one-pot polyol procedure, using the triblock copolymer poly(ethylene glycol)-block-poly(propylene glycol)-blockpoly(ethylene glycol) (PEO-PPO-PEO) as the surfactant. The structure and morphology of the composite nanoparticles were analyzed by X-ray diffraction (XRD), energy dispersive X-ray analysis (EDX), selected area electron diffraction (SAED), a transmission electron microscope (TEM) and high resolution transmission electron microscopy (HRTEM). The characterization showed that the AuZnO composite nanoparticles were spherical, with narrow particle size distribution and high crystallinity. The Fourier transform infrared spectroscopy (FTIR) study confirms the PEO-PPO-PEO molecules on the surface of the composite nanoparticles. The investigations by ultraviolet-visible light absorbance spectrometer (UV-Vis) and photoluminescence spectrophotometer (PL) demonstrate well the dispersibility and excellent optical performance of the AuZnO composite nanoparticles. Photocatalytic activity and reusability of the AuZnO nanoparticles in UV and visible light regions was evaluated by the photocatalytic degradation of Rhodamine B (RhB). The experimental results show that the AuZnO composite nanoparticles with a suitable amount of Au loading have stability and improved photocatalytic activity. AuZnO composite nanoparticles are effective and stable for the degradation of organic pollutants in aqueous solution.

**Keywords:** one-pot polyol synthesis; AuZnO composite nanoparticles; optical performance; photocatalytic activity

## 1. Introduction

Emissions of large amounts of industrial organic wastewater and domestic sewage have resulted in serious surface and groundwater pollution. Water pollution not only poses a great threat to the environment and people's health, but also aggravates the shortage of water resources [1]. In order to deal with the organic pollutants in the water and obtain more water resources, many physical and chemical methods have been tried, such as coagulation/flocculation, chemical oxidation, electrocoagulation, coagulation/carbon adsorption process, etc. [2–11]. Compared with these methods, photocatalytic degradation of organic pollutants is considered to be the most promising technology.

The use of heterogeneous catalyst (semiconductor materials) photocatalytic technology aroused widespread concern, due to the effective and rapid elimination of organic compounds and the relatively low cost [12]. Nowadays, many kinds of semiconductor materials have been studied as photocatalysts, including $TiO_2$, ZnO, $WO_3$, CdS, and so on [13–17]. Among the semiconductor materials, ZnO has the

advantages of wide band gap energy (3.37 eV), high sensitivity, high thermal stability, easy preparation, and low cost, and has been widely used in various fields, such as sensors, electronics, solar cells, and optoelectronics [18–22]. However, its applications are somewhat restricted [23]. Research shows that doping ZnO with the appropriate amount of precious metal can effectively improve its photocatalytic activity [23–26].

We previously reported the one-pot polyol synthesis of poly(ethylene glycol)-block-poly(propylene glycol)-blockpoly(ethylene glycol) (PEO-PPO-PEO)-coated AuZnO composite nanoparticles, using PEO-PPO-PEO molecules as the surfactant [27]. The AuZnO composite nanoparticles have two absorption bands, one centered at 350~360 nm from the band edge of ZnO, and the other at a maximum of 520~550 nm from the surface plasmon resonance (SPR) of the nanostructured Au. The wide optical absorbance of AuZnO composite nanoparticles is promising for photocatalysis, photodegradation, biosensing, and optoelectronic devices. In this work, the AuZnO composite nanoparticles with different compositions were prepared. The characterization shows that the nanoparticles are monosized and have high crystallinity, manifesting excellent dispersibility and optical performance in both organic and aqueous media. The photocatalytic activity of AuZnO composite nanoparticles with different composition were investigated by the photodegradation of RhB under UV and visible light irradiation, and the reusability of AuZnO catalyst is also studied. The results reveal that the AuZnO composite nanoparticles with proper Au loading can improve photocatalytic activity and stability, and are expected to be applied in the field of photocatalysis.

## 2. Experimental

### 2.1. Materials and Reagents

Gold(III)acetate (99.9%), Zinc(II)acetylacetonate (99.9%), PEO-PPO-PEO (Mw = 5800), octyl ether (99%), and 1,2-hexadecanediol (90%) were obtained from J and K Scientific Ltd (Beijing, China). Other reagents include hexane and ethanol. All materials were used as they were, without further purification.

### 2.2. Synthesis of PEO-PPO-PEO-Coated AuZnO Composite Nanoparticles

As illustrated in Scheme 1, the PEO-PPO-PEO-coated AuZnO composite nanoparticles were synthesized according to the methods previously reported in the literature [27]. The synthesis was carried out in a 100 mL three-necked flask-for instance, the AuZnO sample coded as S1 was obtained by mixing $Au(OOCCH_3)_3$ 0.0112 g (0.03 mmol) and $Zn(acac)_2$ 0.3875 g (1.47 mmol) in 10 mL octyl ether with 1,2-hexadecanediol 0.4851 g, and PEO-PPO-PEO 0.7878 g under vigorous stirring. The reaction mixture was first heated to 125 °C in 2 h and held at 125 °C for 1 h. The temperature was then rapidly warmed to 280 °C in 15 min and held at 280 °C for 1 h. After cooling to room temperature, the purple black product was obtained by centrifugation. The product was washed several times with ethanol/hexane at a volume ratio of 2:1, and dried to obtain a sample of about 88 mg. The other two AuZnO samples were prepared analogously, except for the various amounts of precursor, reductant, and surfactant. The AuZnO sample coded as S2 had the precursor, reductant, and surfactant amounts of $Au(OOCCH_3)_3$ 0.0281 g (0.075 mmol), $Zn(acac)_2$ 0.3756 g (1.425 mmol), 1,2-hexadecanediol 0.4851 g, and PEO-PPO-PEO 0.7878 g, and about 93 mg of sample was finally obtained, while about 102 mg of sample S3 was obtained from the mixture of $Au(OOCCH_3)_3$ 0.0561 g (0.15 mmol), $Zn(acac)_2$ 0.3559 g (1.35 mmol), 1,2-hexadecanediol 0.4851 g, and PEO-PPO-PEO 0.7878 g. For a comparative investigation, more composite nanoparticles were similarly synthesized alone from $Au(OOCCH_3)_3$ (coded as Au) and $Zn(acac)_2$ (coded as ZnO) separately. The nominal composition was $Au_{0.03}(ZnO)_{1.47}$ for S1, $Au_{0.075}(ZnO)_{1.425}$ for S2, and $Au_{0.15}(ZnO)_{1.35}$ for S3, respectively. For comparison, ZnO nanoparticles were prepared similarly, using only zinc acetylacetonate as the precursor (Figure S3, shown in supplementary materials).

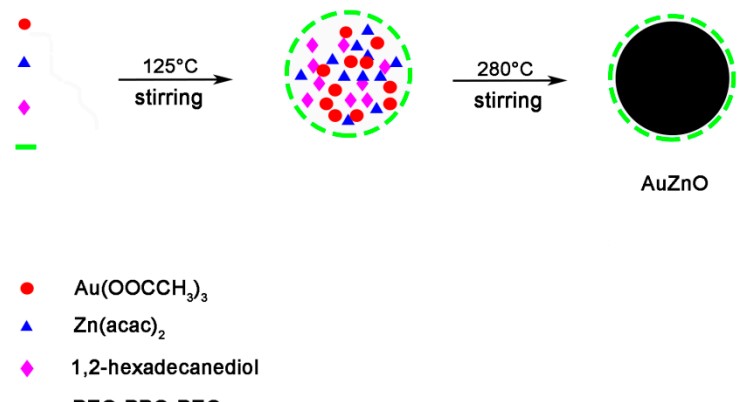

**Scheme 1.** Schematic route for the growth of the AuZnO composite nanoparticles in the nano-micelles formed by the poly(ethylene glycol)-block-poly(propylene glycol)-block-poly(ethylene glycol) (PEO-PPO-PEO) macromolecules.

### 2.3. Characterization

The structure and morphology of the composite nanoparticles were analyzed by XRD (X'Pert Pro $\lambda$ = 1.54056 Å), TEM (JEM-100II), SAED, and EDX, and the elemental compositions were measured by Inductive Coupled Plasma Emission Spectrometer (ICAP6200, Thermo Scientific, Cambridge, UK), in which Au and Zn standard solutions were obtained from A Johnson Matthey Company, and aqua regia was used to dissolve the AuZnO composite nanoparticles. The optical properties of composite nanoparticles were characterized by a UV-visible spectrophotometer (UV-Vis near IR spectrophotometer, Hitachi U4100, Tokyo, Japan) and a photoluminescence (PL) spectrophotometer (Hitachi F7000, Tokyo, Japan). Moreover, the AuZnO composite nanoparticles and pure PEO-PPO-PEO polymer were examined by FTIR using an Avatar 360 FTIR spectrometer (Nicolet Company, Madison, WI, US). X-ray photoelectron spectroscopy (XPS) was carried out on a Thermo ESCALAB250XI photoelectron spectrometer (ThermoFisher Scientific, Waltham, MA, United States) with Al K$\alpha$ X-ray as the excitation source. The thermogravimetric analysis (TGA) was performed using a thermogravimeter (NETZSCH STA449F5, Netzsch, Gemerny). The heating rate was programmed at 10 K·min$^{-1}$, with the protecting stream of N$_2$ flow of 40 mL·min$^{-1}$.

### 2.4. Measurement of Photocatalytic Activity

For photocatalytic studies, a 36 W UV lamp, fired mainly at 365 nm (Philips, Amsterdam, Holland), and a 50 W HSX-F/UV visible output of 390-770 nm (NBeT) was used as light source, and the distance to the reaction beaker was found to be 10 cm. RhB at a concentration of 5 mg/L was prepared by dissolving the dye. A total of 10 mg of the catalyst was added to 30 mL of 5 mg/L RhB in water to carry out the reaction. The mixtures were stirred under dark conditions for 30 min before irradiation, to disperse the catalysts and reach the adsorption equilibrium. Then, under ultraviolet or visible radiation, approximately 4 mL samples were withdrawn every 15 min and separated by centrifuging. The residual concentration of RhB was determined by taking the UV-Vis absorption spectra (TU-1900, Beijing, China), to measure the absorption of RhB in the range of 200~800 nm. The *y*-axis of degradation is called as $C/C_0$, where $C$ is the RhB concentration at each irradiation time interval determined at $\lambda$max, and $C_0$ is the initial concentration when the adsorption-desorption equilibrium was achieved. JCPDS (joint committee on powder diffraction standards).

## 3. Results and Discussion

### 3.1. Transmission Electron Microscope Morphology and Nanostructures of AuZnO Composite Nanoparticles

The morphology and particle size distribution of AuZnO composite nanoparticles(S1) were analyzed by TEM. As showed in the Figure 1a, the composite nanoparticles are highly crystalline, uniform, and nearly spherical. The histogram was obtained by counting a series of TEM images of the composite nanoparticles using Nano measurer software, as shown in the illustration, revealing that the composite nanoparticles have a narrow particle size distribution and are consistent with the Gaussian distribution with a particle size of ~14.5 (±0.9) nm. The TEM analysis gives the particle size of ~13.8 (±1.4) nm and ~15.3 (±1.5) nm in diameter for S2 and S3, respectively (shown in supplementary materials, Figures S1 and S2). Figure 1b shows the HRTEM image of a single AuZnO composite nanoparticle(S1). Obviously, highly regular lattices are uniformly distributed on the composite nanoparticle, as labeled, with the spacing of 2.35 Å indicating the projection of the Au (111) plane, whereas the spacing of 2.60 Å corresponds to the ZnO (002) plane. The result shows that Au and Zn are present in the same composite nanoparticle. Figure 1c shows a typical TEM-EDX point-detection for AuZnO composite nanoparticles(S1), which clearly shows that the composite nanoparticles contain zinc and gold elements. As shown in Figure 1d, the typical SAED pattern with bright circular rings corresponding to the (100), (101), (200), (110), (103) and (220) planes show that the nanoparticles obtained are highly crystalline. The sharpness and multi-diffraction spots observed in the pattern confirm the high crystallinity of AuZnO composite nanoparticles. As shown in Figure S6, further TEM element mapping provides more solid information for AuZnO(S1) composite nanoparticles; the positions of the composite nanoparticles as recorded by the elemental mapping in Figure S6b for Au and Figure S6c for Zn, which are consistent with those in the bright-field image of Figure S6a (Figure S6, shown in supplementary materials). The results confirmed the synthesis of AuZnO composite nanoparticles.

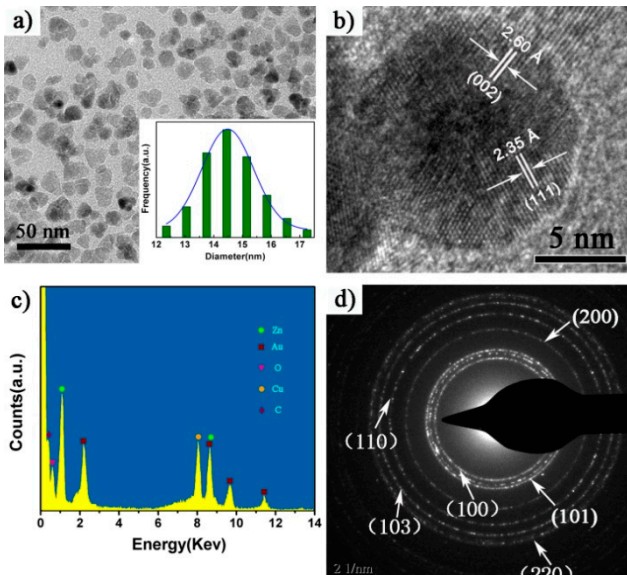

**Figure 1.** Transmission electron microscope (TEM) image of AuZnO composite nanoparticles(S1). (**a**) Bright-field TEM particle size distribution (histogram) with Gaussian function fit (in curve) of the composite nanoparticles; (**b**) HRTEM of a single nanoparticle; (**c**) point-detection energy dispersive X-ray analysis (EDX) analysis chart; (**d**) selected-area electron diffraction pattern of the nanoparticles.

The formation of the AuZnO composite nanoparticles(S1–S3) was further confirmed by X-ray crystal structure analysis. As shown in Figure 2a–c, the diffraction peaks of the AuZnO composite nanoparticles may be indexed to two sets, one in the inverted triangles, corresponding to the Au

positions of the (111), (200), (220), (311), and (222) planes (JCPDS NO. 65-8601), and the other in the squares corresponding to the ZnO positions of the (100), (002), (101), (110), (103), (200), (112), (201), and (004) planes (JCPDS NO. 05-0664), indicating the presence of Au and ZnO. The intensity of the diffraction peaks of Au become weakened from S3 (Figure 2a) to S2 (Figure 2b) to S1 (Figure 2c), because Au amount decreases from S3 ($Au_{0.15}(ZnO)_{1.35}$) to S2 ($Au_{0.075}(ZnO)_{1.425}$) to S1 ($Au_{0.03}(ZnO)_{1.47}$). In contrast, the diffraction of the reflective ZnO material is greatly enhanced. The average particle size of the composite nanoparticles is ~14.1 nm, ~13.4 nm, and ~15.1 nm for S1, S2, and S3, respectively, by the Scherrer equation (Equation (1)), based on the full width at half maximum (FWHM); this is slightly smaller than the statistical count of the above TEM analysis, probably due to the small size effect of the nanoparticles, leading to the broadening of the XRD diffraction peaks [28].

$$D_{hkl} = \frac{180° \kappa \lambda}{\pi B_{1/2} \cos \theta} \tag{1}$$

where $\lambda$ is the X-ray wavelength (1.54 Å), $\theta$ is the Bragg diffraction angle, $k = 0.89$, and $B_{1/2}$ is the full width at half maximum.

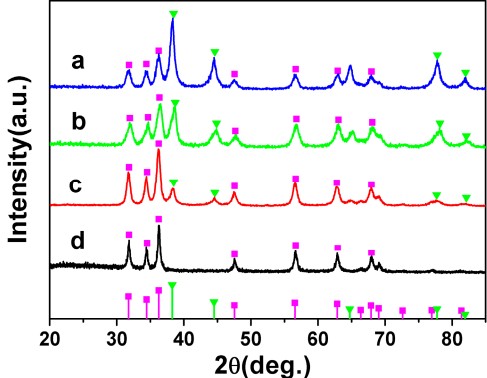

**Figure 2.** X-ray diffraction (XRD) patterns for the AuZnO composite nanoparticles, (**a**) S3, (**b**) S2, (**c**) S1, and (**d**) ZnO, with a bar diagram for the JCPDS of ZnO (squares) and Au (inverted triangles).

The XPS analysis was carried out to investigate the chemical composition of the AuZnO nanoparticles, and the corresponding experimental results are shown in Figure 3. The binding energy in the XPS spectra was calibrated using C 1s (284.8 eV). There are no peaks for other elements, except for Zn, O, Au, and C observed from the full XPS spectra of Figure 3a. The presence of carbon comes largely from the surfactant (PEO-PPO-PEO) molecules on the surface of the resulting nanoparticles. Therefore, it is concluded that the nanoparticle is mostly composed of three elements: Zn, Au, and O. In Figure 3a, the composite nanoparticles display a doublet at about 1022 and 1045 eV, corresponding to the Zn $2p_{3/2}$ and $2p_{1/2}$ core levels [29]. The first peak is attributed to $Zn^{2+}$ ions in the oxygen-deficient ZnO matrix. Moreover, all of the Zn $2p_{3/2}$ XPS peaks are sharp. Thus, it can be confirmed, that a Zn element exists mainly in the form of $Zn^{2+}$ on the samples' surfaces [30]. The peaks observed at 83.2 and 87.3 eV (Figure 3b) correspond to the Au $4f_{5/2}$ and Au $4f_{7/2}$ states of the metallic Au. The binding energy of the Au 4f states is shifted obviously to lower values, as observed for other AuZnO composites when compared to bulk Au (from 84.0 and 87.7 eV, respectively [31]). This effect is often ascribed to the fact that interaction between Au and ZnO-namely, the electron transfer-takes place between Au and ZnO at the interfaces where the Au and ZnO come into contact [31]. Thus, the analysis illustrates that the AuZnO nanoparticles are dominantly comprised of Au and ZnO, which is in agreement with the XRD and TEM results, as previously addressed.

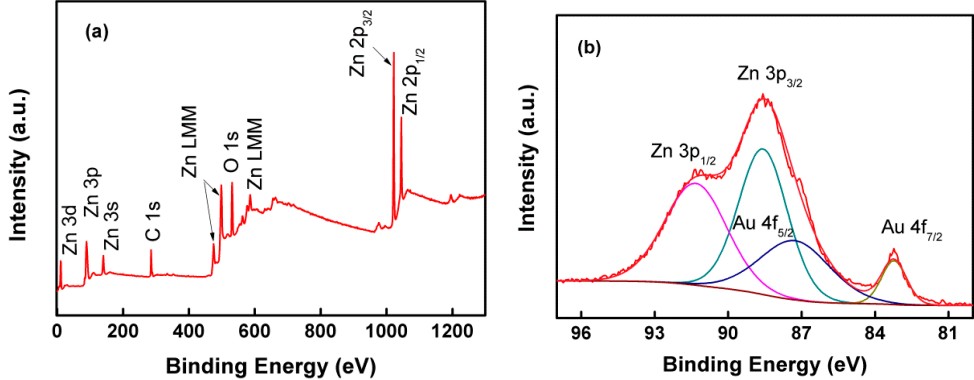

**Figure 3.** X-ray photoelectron spectroscopy (XPS) spectra of the AuZnO(S2) composite nanoparticles. (**a**) Surveying XPS spectrum, and (**b**) Zn 3p and Au 4f spectrum.

The elemental content of the AuZnO composite nanoparticles was tested by ICAP analysis. The results show that Au element content is $7.0 \pm 0.3\%$, $13.1 \pm 1\%$, and $26.1 \pm 1\%$ for S1, S2, and S3, respectively, while the Zn element content is $63.0 \pm 0.3\%$, $55.1 \pm 0.7\%$, and $45.0 \pm 1.3\%$ for S1, S2, and S3, respectively.

The spectra of the purified AuZnO composite nanoparticles and pure PEO-PPO-PEO macromolecules were analyzed by FTIR, indicating that PEO-PPO-PEO macromolecules are present on the surface of the AuZnO composite nanoparticles [32]. Figure 4 compares the FTIR spectra of purified AuZnO composite nanoparticles and pure PEO-PPO-PEO molecules. In Figure 4d, a strong characteristic peak at about 1122 cm$^{-1}$ for pure PEO-PPO-PEO molecules belongs to the C-O-C stretching vibration, which is usually between 1250 cm$^{-1}$ and 1000 cm$^{-1}$, with a characteristic peak at ~1475 cm$^{-1}$ that is due to C-H bending vibration [32]. These characteristic vibration and bending modes are reproduced in the FTIR spectra of the AuZnO composite nanoparticles, but the C-O-C stretching vibration red shifts to the positions of ~1116 cm$^{-1}$, ~1118 cm$^{-1}$, and ~1123 cm$^{-1}$ for S1, S2, and S3 composite nanoparticles (Figure 4a–c), and the C-H bending vibration shifts to ~1589 cm$^{-1}$. In addition, the ribbon shape and location are significantly different for AuZnO composite nanoparticles than for pure PEO-PPO-PEO molecules, due to changes in the elastic constants that are the result of the great curvature and strong interaction. The blue shift and change in the band shape of the C-O-C stretching and C-H bending bands may involve the coordination of the oxygen atoms in the main chains of PEO-PPO-PEO to the metallic atoms [32–34]. Still, the characteristic band of Zn-O in the spectra of the PEO-PPO-PEO-coated AuZnO composite nanoparticles is located at ~443 cm$^{-1}$, ~461 cm$^{-1}$, and ~457 cm$^{-1}$ for AuZnO(S1) (Figure 4a–c), which is definitely distinct from the absence of absorption in the corresponding position in the spectra of the pure PEO-PPO-PEO molecules (Figure 4d). When excess PEO-PPO-PEO molecules were removed by purification, the analysis showed that PEO-PPO-PEO molecules were coated on the surface of the AuZnO composite nanoparticles.

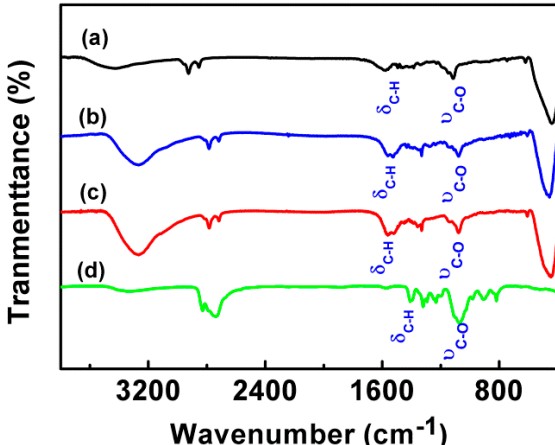

**Figure 4.** Fourier transform infrared spectroscopy (FTIR) spectra of the AuZnO(S1–S3) composite nanoparticles (**a**–**c**) and the PEO-PPO-PEO (**d**).

### 3.2. Optical Properties of AuZnO Composite Nanoparticles

As mentioned above, the PEO-PPO-PEO molecules are placed on the surfaces of AuZnO composite nanoparticles. Therefore, the composite nanoparticles are both hydrophobic and hydrophilic, and their transfer between the non-polar and the polar solvents does not require further surface modification. The UV-Vis absorption and fluorescence spectra of the AuZnO composite nanoparticles dispersed in water for the former and in hexane for the latter were recorded in Figures 5 and 6, respectively. The UV-Vis absorption spectra of AuZnO composite nanoparticles are compared with Au and ZnO nanoparticles of similar size dispersed in water, as shown in Figure 5. The spectra clearly show band edges at 364 nm, 358 nm, and 362 nm for S1, S2, and S3 respectively, with similar shapes but blue shifted bands compared to the pure ZnO (366 nm), which is the most characteristic absorption of the ZnO semiconductor. The shift and band shape change are related to the SPR of the Au. For S3, there are two kinds of absorption bands. One is centered at 362 nm from the band edge of ZnO, and the other is centered with a maximum at 569 nm from the SPR of the nanostructured Au. Compared to pure Au nanoparticles (542 nm), the SPR band shifts to higher wavelengths. Similar SPR band red shift phenomena were reported for Au-ZnO, AuFe, Ag-ZnO, and Ag-Fe$_3$O$_4$ nanocomposites [35–38]. These shifts may be due to the charge transfer between the Au nanoparticles and the ZnO nanoparticles in the heterodimeric structure [39]. The AuZnO heterodimeric nanostructures that regulate SPR band red shift can open the door to advanced materials for construction.

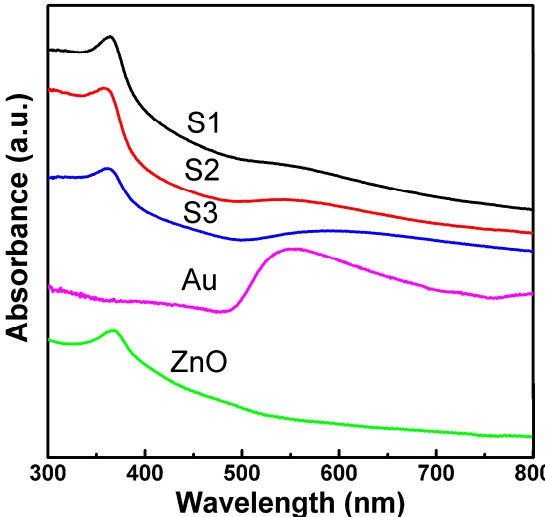

**Figure 5.** Ultraviolet-visible light absorbance spectrometer (UV-Vis) spectra of Au, ZnO, and AuZnO nanoparticles (S1–S3) dispersed in water.

The photoluminescence spectrum of AuZnO composite nanoparticles in *n*-hexane was excited at a wavelength of $\lambda = 360$ nm, as shown in Figure 6. Emissions come from ZnO and Au nanostructures. In the case of sample S1, a unique emission of 400 nm appears, as shown by the inverted triangle, with a distinct peak at about 474 nm and a relatively strong emission at about 576 nm. In the spectrum, the blue band around 400 nm is most likely to undergo acceptor levels from the donor level of the Zn gap to the Zn vacancies [40]. The blue band at about 474 nm is due to the electronic transition of the donor level produced by valence band oxygen vacancies [41], and emission at about 580 nm is generally attributed to singly ionized oxygen vacancies in ZnO, due to recombination between electrons in the deep or superficial defect levels and holes in the valence band [42]. In sample S2, the intensity of the band gap emission is enhanced, and the position is slightly shifted to 402 nm. The same is true for the distinct plateau at 474 nm, which also has a strong emission at about 578 nm. Sample S3 also exhibited a strong emission at about 403 nm, with a firm plateau at about 476 nm and another emission at about 580 nm. It is of interest to emphasize that the photoluminescence spectra of the polymer-coated AuZnO composite nanoparticles (S1–S3) dispersed in hexane is gradually red shifted with the increase of concentration of Au. This is because that when nanosized Au combines with ZnO, the interface interaction causes the influence of the charge change on the Au surface [43].

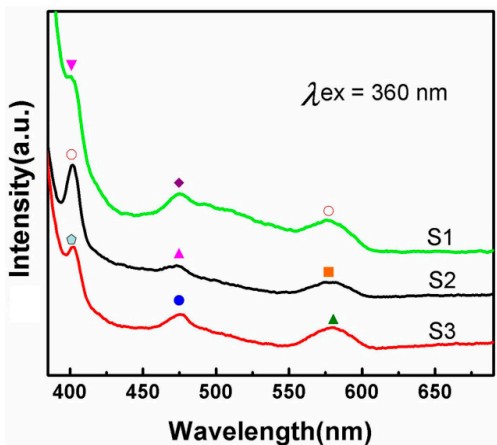

**Figure 6.** Photoluminescence spectra of AuZnO(S1–S3) in *n*-hexane under excitation of the wavelength $\lambda = 360$ nm.

Figure S7 shows the thermogravimetric and differential thermal analysis (TG-DTA) curves of PEO-PPO-PEO-terminated AuZnO(S2) composite nanoparticles (Figure S7, shown in supplementary materials). It can be seen from the TG curve in the figure that the weight loss of PEO-PPO-PEO-capped AuZnO composite nanoparticles can be divided into three stages. From room temperature to 260 °C, weight loss of PEO-PPO-PEO-terminated AuZnO composite nanoparticles is 2%, mainly due to the removal of physically adsorbed water and chemisorption water [44]. The second stage weight loss is about 4%, corresponding to the volatilization and decomposition of PEO-PPO-PEO molecules adsorbed on the surface of the composite nanoparticles (between about 260 and 450 °C) [45]. In the third stage, there is no obvious weight loss. The differential thermal analysis (DTA) of samples was shown in Figure S7. Three endothermal peaks 70 °C, 178 °C, and 413 °C correspond well to the desorption of physically adsorbed water, removal of chemically adsorbed water, and the volatilization and decomposition of PEO-PPO-PEO molecules adsorbed on the surface of the composite nanoparticles, respectively. On the DTA curve, a main exothermic effect was observed between 460 and 600 °C, with a maximum at about 544 °C, indicating that the thermal events can be associated with the burnout of organic species involved in the precursor powders (organic mass remained from PEO-PPO-PEO) of the residual carbon, or due to direct crystallization of AuZnO nanocrystalline from the amorphous component [46]. The TG-DTA curves of PEO-PPO-PEO-terminated AuZnO composite nanoparticles (S1–S3) are similar.

### 3.3. Photocatalytic Testing

The photocatalytic performances in UV and visible light regions were investigated via the degradation of RhB, which is commonly used as a probe molecule to evaluate the photocatalytic activity of catalysts. Figure 7 shows the photocatalytic activity of prepared AuZnO(S1–S3) and ZnO for the degradation of RhB under (a) UV irradiation and (b) visible light irradiation. Experiments without a catalyst addition and with no light conditions were also shown in the figure for comparison purposes. Figure 7a presents a comparative study for the degradation of RhB under UV in the presence of different catalysts. RhB decomposes 85.7% (curve 1), 57.2% (curve 2), and 34.4% (curve 3), respectively, when AuZnO(S1–S3) acted as a photocatalyst for 90 min under UV irradiation, while 53.1% (curve 4), and 6.4% (curve 6) degradation were observed in the presence of ZnO under UV irradiation and in the dark. At the same time, 4.3% (curve 5) is barely exposed, and only with UV light (photolysis). Figure 7b shows the photocatalytic activity of the prepared AuZnO(S1–S3) and ZnO, to degrade RhB under visible light irradiation. It is observed that the degradation of RhB is 63.5% (curve 1), 31.8% (curve 2), 18.7% (curve 30), and 49.9% (curve 4) respectively. The photocatalytic activity order of AuZnO composite, both under UV and under visible light is S3 < S2 < S1. Samples S1 and S2 reveal better capability than pure ZnO under UV irradiation, while only S1 shows better capability than ZnO under visible light. Sample S3 shows the lowest photocatalytic activity. XRD analysis indicates that Au amount increased from S1, S2, to S3, revealing the surfaces of S3 could almost completely covered by Au, which is most likely the main reason for the poor performance of S3 owing to the reduced the surface contact area between RhB and ZnO. In this experimental range, the smaller amount of Au will make the better photocatalytic activity of AuZnO composite nanoparticles. We believe that the appropriate amount of Au on ZnO is the main factor for the enhancement of the photocatalytic activity.

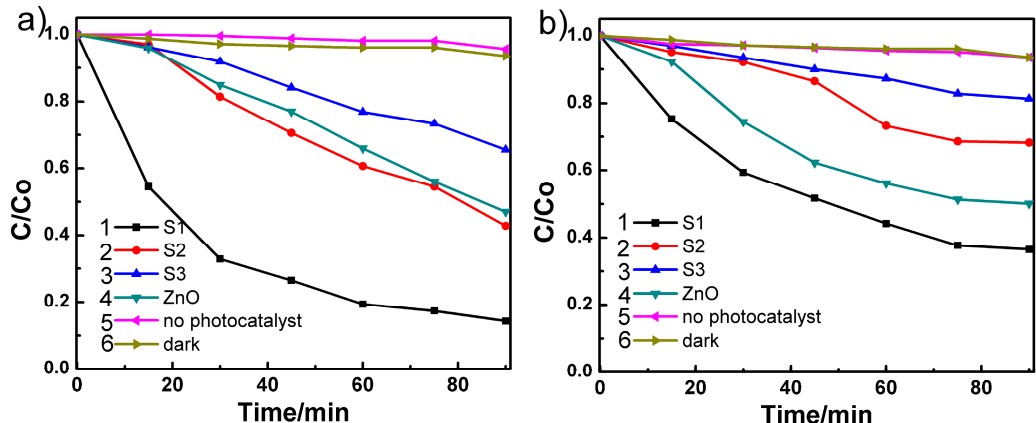

**Figure 7.** Photocatalytic degradation of RhB by AuZnO composite nanoparticles(S1–S3) and ZnO. (**a**) UV irradiation and (**b**) visible light irradiation.

In order to explore the reusability of the AuZnO catalyst under UV and visible light conditions, RhB was degraded with the AuZnO catalyst and circulated under the same conditions for three cycles, and the results are shown in the Figure 8. It was noteworthy that the AuZnO(S1) photocatalyst still showed good photostability and photocatalytic activity under UV (Figure 8a) and visible light irradiation (Figure 8b), even after repeated use for three cycles. The recyclability of AuZnO is due to the resistance and stability to photo-corrosion, which plays a pivotal role for its efficient use, especially for a greener and environmentally-friendly approach. Similarly, the recyclability of AuZnO(S2-S3) composite nanoparticles was tested under UV irradiation (Figure S4). The photocatalytic activities of the S1 composite nanoparticles were 85.7%, 90.9%, and 86.2%, those of S2 were 57.2%, 62.9%, and 63.4%, and those of S3 were 34.4%, 36.3%, and 33.0%. The results show that the photocatalytic activity of AuZnO(S1–S3) remains unchanged after repeated use for three cycles. The photocatalytic activities order of AuZnO composite under UV irradiation is always S3 < S2 < S1. The AuZnO(S3) composite nanoparticles after cyclic experiments were characterized by XRD. As showed in Figure S5a, the structure of AuZnO(S3) composite nanoparticles did not change after cycling experiments, compared with the AuZnO(S3) composite nanoparticles of Figure S5b (Figure S5, shown in supplementary materials). This indicates that the AuZnO(S3) composite nanoparticles have good stability as a catalyst in the dye solution.

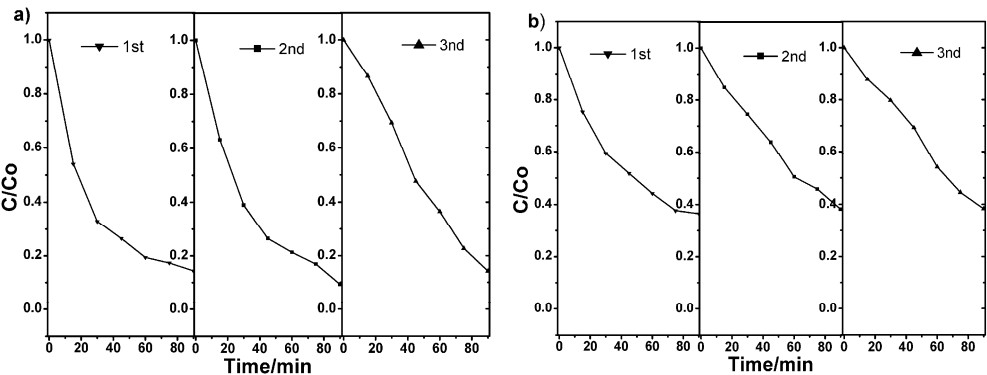

**Figure 8.** Degradation of RhB by AuZnO(S1) catalyst for three cycles. (**a**) UV and (**b**) visible light irradiation.

### 3.4. Mechanisms in Enhancing Photocatalytic Activity

Figure 9 shows the photocatalytic degradation of RhB routes of the synthesized AuZnO composite nanoparticles. Under dark conditions, the Au surface electrons of the AuZnO composite nanoparticles

in the RhB solution can be transferred to the dye [47]. As showed in Figure 9a, the electrons move from the excited RhB (RhB*) to the conduction band of ZnO under visible light irradiation, and then electrons are transferred from the ZnO surface to the surface of Au [48]. In this system, Au acts as a capture site for photoexcited electrons, and promotes the separation of photogenerated electron-RhB pairs, thereby improving the photocatalytic activity of the AuZnO photocatalyst. While the catalyst is irradiated with UV light, the photon energy is greater than or equal to the energy of the ZnO nanoparticles, which facilitates the transfer of photoelectrons from the valence band (VB) to the conduction band (CB), while leaving the same number of holes in the VB. As showed in Figure 9b, since the energy level at the bottom of the CB is higher than the Fermi level of AuZnO composite nanoparticles, the potential energy generated can promote the transfer of photoelectrons from ZnO to Au. The photoelectrons can be trapped by $O_2$ on the ZnO surface to form superoxide anion radicals ($\cdot O^{2-}$). At the same time, $H_2O$ can trap holes to form hydroxyl radical $\cdot OH$, which is a kind of strong oxidizer, and thus oxidizes dye molecules into $CO_2$, $H_2O$, etc. [49].

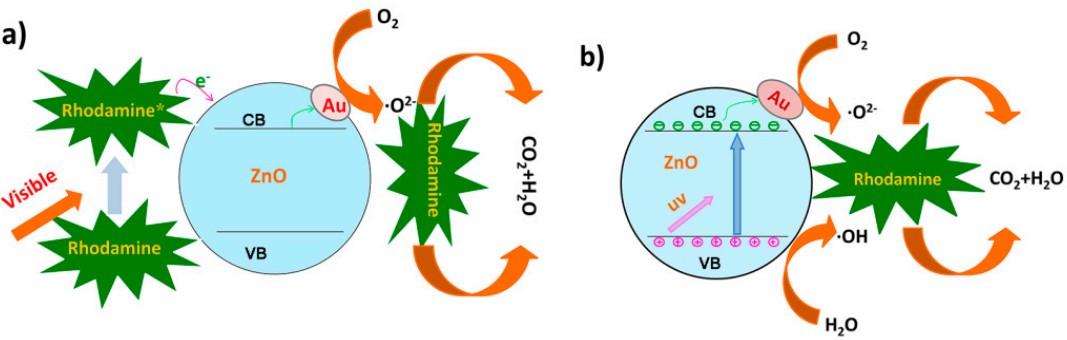

**Figure 9.** Photocatalytic mechanisms for RhB degradation by AuZnO composite nanoparticles. (**a**) AuZnO composite nanoparticle degradation of RhB under visible light irradiation, and (**b**) AuZnO composite nanoparticle degradation of RhB under UV irradiation.

### 4. Conclusions

In conclusion, we propose a simple method to prepare AuZnO composite nanoparticles coated with PEO-PPO-PEO, which has better photocatalytic efficiency than pure ZnO under appropriate Au loading. The FTIR evaluation showed that PEO-PPO-PEO molecules are present on the surface of the AuZnO composite nanoparticles. The structural and morphological analysis showed that the composite nanoparticles have a narrow particle size distribution and high crystallinity. UV-Vis and PL analyses show that AuZnO composite nanoparticles have good dispersibility and excellent optical properties, which are beneficial for photocatalytic activity. The evaluation of the photocatalytic activity of AuZnO composite nanoparticles of different compositions is achieved by degrading RhB under UV and visible light irradiation conditions, revealing that the AuZnO composite nanoparticles with proper Au loading possess enhanced photocatalytic activity. The photocatalytic activities order of AuZnO composite nanoparticles under both UV and visible light is S3 < S2 < S1. Samples S1 and S2 reveal better capability than pure ZnO under UV irradiation, while only S1 shows better capability than ZnO under visible light. The photocatalytic efficiency of the AuZnO composite nanoparticles (S1–S3) was found to be unaltered after three cycles of use under UV irradiation, revealing that the PEO-PPO-PEO-coated AuZnO composite nanoparticles are stable and environmentally friendlier and greener.

**Supplementary Materials:** The following are available online at http://www.mdpi.com/2076-3417/9/1/111/s1, Figure S1: TEM image of AuZnO composite nanoparticles (S2), (a) Bright-field TEM, particle size distribution (histogram) with Gaussian function fit (in curve) of the composite nanoparticles, (b) HRTEM of single nanoparticle, (c) Point-detection EDX analysis chart, (d) selected-area electron diffraction pattern of the composite nanoparticles, Figure S2: TEM image of AuZnO composite nanoparticles (S3), (a) Bright-field TEM, particle size distribution (histogram) with Gaussian function fit (in curve) of the composite nanoparticles, (b) HRTEM of single nanoparticle,

(c) Point-detection EDX analysis chart, (d) selected-area electron diffraction pattern of the composite nanoparticles, Figure S3: TEM image of ZnO nanoparticles, (a) Bright-field TEM, particle size distribution (histogram) with Gaussian function fit (in curve) of the composite nanoparticles, (b) HRTEM of single nanoparticle, Figure S4: Degradation of RhB by AuZnO composite nanoparticles for 3 cycles under UV irradiation (a) S2, (b) S3, Figure S5: XRD patterns for the AuZnO composite nanoparticles, (a) S3 after the cycle test, (b) S3, inverted triangles indicate the positions of Au peaks, and squares indicate the positions of ZnO peaks, Figure S6: TEM elemental mapping analysis of the AuZnO(S1) composite nanoparticles, (a) Bright field imaging, (b) O elemental mapping, (c) Au elemental mapping, (d) Zn elemental mapping, Figure S7: TG-DTA analysis of PEO-PPO-PEO coated AuZnO(S2) composite nanoparticles from 25 to 600 °C.

**Author Contributions:** Conceptualization, C.M., X.W., and H.L.; data curation, C.M., X.W., and S.Z.; formal analysis, C.M., X.W., S.Z., X.L. (Xuemei Li), X.L. (Xiao Liu), and H.L.; investigation, Y.C.; resources, X.L. (Xiao Liu), Y.C. and R.X.; software, X.L. (Xuemei Li); writing-original draft, C.M. and X.W.; writing-review and editing, H.L.

**Funding:** This work was supported in part by the National Natural Science Foundation of China (No. 51172064), and the Key Scientific Research Projects of Henan Province Colleges and Universities, Foundation of Education Department of Henan Province, China (No. 16A150002).

**Conflicts of Interest:** The authors declare no conflict of interest.

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
