# Peer review of "Photocatalytic Activity of Monosized AuZnO Composite Nanoparticles"

_applsci, doi:10.3390/app9010111_

Round 1
Reviewer 1 Report
Applied Sciences-MDPI
Manuscript ID: applsci-389879
Manuscript Type: article
Title: “Photocatalytic activity of monosized AuZnO composite nanoparticles”
Authors: Chenguang Ma, Xianhong Wang, Shixia Zhan, Xuemei Li, Xiao Liu, Yun Chai, Ruimin Xing, Hongling Liu
The manuscript by Liu et al describes the synthesis, characterization and photocatalytic activity of AuZnO composite nanoparticles. After careful evaluation of this manuscript, I inform you that this manuscript may be suitable for publication in Applied Sciences after a major and thorough revision. Here are some questions that must be carefully considered and addressed.
1) The authors starting from the title, often use the term “monosized”, but it is not clear to me what they do really mean with this term, also in consideration of the fact that the synthetic protocol used is described in reference 21 to give ZnO-Au hybrid nanoparticles. Please explain.
2) The manuscript is not easy to understand and read. One reason is that the English is truly not satisfactory. The manuscript contains a very large number of typographical and grammatical errors. throughout the manuscript/supplementary and a robust language revision is absolutely needed. The second reason is the total lack of a scheme describing the formation of the AuZnO composite material.
3) In the introduction, line 36, one or more references must be added.
4) Regarding the characterization NO information is given about the instrumentation used, please provide. More serious characterization needs to be done also in consideration of the photocatalytic application. The amount of gold and Zinc in the composites must be confirmed by AAS or ICP analyses and not simply calculated on the quantities used in the fabrication protocol. This is important also in consideration that in paragraph 3.3, line 223, out of the blue the authors write: “XRD analysis indicates that Au amount increased from S1, S2 and S3, revealing the surfaces of S3 could almost covered by Au”. To this respect a XPS analysis would be nice in order to establish the surface compositions.
The adsorption of the copolymer PEO-PPO-PEO PVP on the samples must be quantitatively measured by thermogravimetric (TGA) analyses.
Regarding the TEM analyses after enlarging the images the nanoparticles appear to me all but spherical and a more serious description should be done. Concerning the histogram, the authors must say on how many particles the analyses was made, what kind of software they used and the error made in the measurement. Also add in the supplementary a Table with all the SAED reflection indexes.
In the XRD part the authors describe the average particle size obtained from the Scherrer equation with three digits (i.e. 11.5 nm) using the symbol of ca. instead of reporting the data with the error as requested in any scientific work! In order to made the manuscript more clear and scientifically acceptable, I suggest to make a table summarizing the main dimensional features obtained using TEM and XRD (with errors). With regard to the XRD spectra a table also reporting the 2Ɵ diffraction angles must be added in the manuscript or in the supplementary in order to help the comparison with the JPCDS reference.
Author Response
We are very grateful to the referee for their positive comments and some suggestions for revision. Please refer to the attachment for details.

Reviewer 2 Report
Please find attached for the comments.

Author Response

(The authors gave the same response as above.)

Reviewer 3 Report
The authors report a technique for preparing AuZnO composite nanoparticles coated with a triblock copolymer (PEO-PPO-PEO). Even if the technique to prepare bare AuZnO is well known in literature, the addition of the external copolymer was already reported by the authors in 2014.
They present an enhanced photocatalytic activity of the obtained nano-composites in solutions of RhB. The paper is well written, presenting only some parts that need to be improved in order to be suitable for publication on Applied Science.
In the following some comments and suggestions to improve the readability of the text:
1) From the very beginning, it is not clear to me the real distribution of all the components in the final PEO-PPO-PEO coated AuZnO composite nanoparticles. In particular, in the literature it is reported about Au@ZnO core/shell nanoparticles. What is the real result of synthesis in this case? What they mean with the word “composite”?
The authors should comment about the final result in terms of distribution of the obtained AuZnO nanoparticles. I am referring, in particular, to the TEM image of figure 1, in which it seems to see an inner core surrounded by a shell. Please comment on this point and include it in the main text to improve the reader clarity.
2) Numbers reported in the samples S1, S2 and S3 as subscript in lines 80-81, or 134, may be used to respond to the previous point, but they are not clear to a reader that is not in the field. The authors should explain how they arrive to these numbers referred to the nominal composition of the obtained nanoparticles.
3) In Figure 2, authors reports some squares and inverted triangles in the bottom of the graph. What they represent? In particular, there are some squares of this bottom part that do not have any counterpart in the top curves (a-d). Please comment on this and improve the figure explanation.
4) Figure 3 contains some issues to be fixed: the term “Tranmenttance” in the vertical axis doesn’t mean nothing. Please correct it. In the same graph, the wave number reported in the x axis are too close to each others. Use a different way to show them for example scientific notation.
5) In line 174 the authors report about the UV-Vis absorption and fluorescence spectra of the AuZnO composite nanoparticles, saying that the samples are dispersed in different solvents, depending on the type of experiment. Why they need to change solvent? Usually Abs and PL experiments are conducted on the same solution, in order to appreciate the Stokes shift. In case of a different solvent, the environment is changed and no comparison be made. Please comment on this point.
6) In Figure 4, the authors report about a blu-shift in the absorption bands of samples S1-S3 with respect to the pure ZnO. I noticed from the graph that the blue shift does not seem to have the same behaviour for S1, S2 and S3? S3, in particular, results red shifted with respect to S2. A grid could help the reader to better follow the real behavior. The authors should explain this behavior.
7) In Figure 8, the sketch present some issues. In an energy scale, usually the conduction band (CB) has a higher energy with respect to the valence band (VB). In the sketch, the authors show an inverted situation, as also reported in the reference 35Is this a mistake or they want to show a different situation? Please explain and comment.
8) The tentative explanation of the photocatalytic degradation of RhB in solution (Sect. 3.4) has to be re-phrased and, possibly, improved. It seems to contain too many lacks.
9) I think that the supplementary info has to be presented with some explanation about the three figures, and cannot be presented only as a list of figures with captions.
10) Line 55, and of the line, “of of”, correct. Line 208, “photcatalytic”, correct.
Author Response

(The authors gave the same response as above.)

Reviewer 4 Report
Reviewer Comments on applsci-389879
Photocatalytic activity of monosized AuZnO composite nanoparticles
In this manuscript, the authors have synthesized monosized AuZnO composite nanoparticles with different compositions and tested for photocatalytic degradation of Rhodamine B (RhB). However, the experimental data were not enough to understand the structure of the catalysts. Thus, I cannot recommend the publication of this paper in the present form.
1) The surface composition and chemical state of the AuZnO composite nanoparticles must be analyzed by XPS.
2) Authors should provide the Au distribution in ZnO matrix by STEM or SEM-EDS color mappings.
3) To understand the superiority of the catalyst, it would be better to characterize the spent catalyst after the cycle test.
4) There are some typos could be found. The authors should correct those errors before publication.
Author Response

(The authors gave the same response as above.)

Round 2
Reviewer 1 Report
Applied Sciences-MDPI
Manuscript ID: applsci-389879
Manuscript Type: article
Title: “Photocatalytic activity of monosized AuZnO composite nanoparticles”
Authors: Chenguang Ma, Xianhong Wang, Shixia Zhan, Xuemei Li, Xiao Liu, Yun Chai, Ruimin Xing, Hongling Liu
The authors have certainly improved, in an amazingly short time, their manuscript. However there is a point that is not convincing me and regards the gold and zinc content. In the Experimental part, paragraph 2.2 entitled “Synthesis of PEO-PPO-PEO-coated AuZnO composite materials” they must declare the weight of the final material obtained after work-up in this way the reader has an idea if the final Au and Zn content later reported is in keeping with the preparation. Moreover they must describe in detail the methodology used to dissolve the samples (not a trivial task) before the ICP analysis and the kind of standards used for the calibration lines. Finally in a scientific, reliable work, the Au content cannot be given with only one digit (6, 8, 29%) and without reporting the error made in the analysis. The Zn content is still missing, why?
Regarding the TEM analyses, the authors should write also in the experimental part of the manuscript (and not only in the cover letter) on how many particles the analyses was made, what kind of software they used and the error made in the measurements.
Regarding the TGA (and not TAG!) the authors write TGA curves but I see only one curve for S2. Where are the others? Are they similar? Please also report the first derivative of the curve in order to better individuate the weight losses.
In general, as evident reading any scientific book, figures must be embedded within the corresponding text. In other word the authors should write first a line or two of text and then the corresponding figures not the contrary
Author Response
Reply to Referees’ comments
We appreciate the Referees’ positive and useful comments, and address them one by one as follows:
Reviewer #1
Reviewer number:1
1. The authors have certainly improved, in an amazingly short time, their manuscript. However there is a point that is not convincing me and regards the gold and zinc content. In the Experimental part, paragraph 2.2 entitled “Synthesis of PEO-PPO-PEO-coated AuZnO composite materials” they must declare the weight of the final material obtained after work-up in this way the reader has an idea if the final Au and Zn content later reported is in keeping with the preparation. Moreover they must describe in detail the methodology used to dissolve the samples (not a trivial task) before the ICP analysis and the kind of standards used for the calibration lines. Finally in a scientific, reliable work, the Au content cannot be given with only one digit (6, 8, 29%) and without reporting the error made in the analysis. The Zn content is still missing, why?
Thank you very much for your positive comments and suggestions. Per the Referee’s constructive comments, the manuscript has been modified as follows.
The synthesis was carried out in a 100 mL three-necked flask, for instance, the AuZnO sample coded as S1 was obtained by mixing Au(OOCCH3)3 0.0112 g (0.03 mmol), Zn(acac)2 0.3875 g (1.47 mmol) in 10 mL octyl ether with 1,2-hexadecanediol 0.4851 g, and PEO-PPO-PEO 0.7878 g under vigorous stirring. The reaction mixture was first heated to 125 °C in 2 h and held at 125 °C for 1 h. The temperature was then rapidly warmed to 280 °C in 15 min and held at 280 °C for 1 h. After cooling to room temperature, the purple black product was obtained by centrifugation. The product was washed several times with ethanol/hexane at a volume ratio of 2:1, and dried to obtain a sample of about 88 mg. The other two AuZnO samples were prepared analogously except for the various amounts of precursor, reductant and surfactant. The AuZnO sample coded as S2 had the precursor, reductant and surfactant amounts of Au(OOCCH3)3 0.0281 g (0.075 mmol), Zn(acac)2 0.3756 g (1.425 mmol), 1,2-hexadecanediol 0.4851 g and PEO-PPO-PEO 0.7878 g, about 93 mg of sample was finally obtained, while about 102 mg of sample S3 was obtained from the mixture of Au(OOCCH3)3 0.0561 g (0.15 mmol), Zn(acac)2 0.3559 g (1.35 mmol), 1,2-hexadecanediol 0.4851 g and PEO-PPO-PEO 0.7878 g.
The elemental compositions were characterized by Inductive Coupled Plasma Emission Spectrometer (Thermo ICAP6200), in which Au and Zn standard solutions were obtained from A Johnson Matthey Company, and aqua regia was used to dissolve the AuZnO composite nanoparticles.
The elemental content of the AuZnO composite nanoparticles was tested by ICP analysis. The results show that Au element content is 7%(±0.3%), 13%(±1%) and 26%(±1%) for S1, S2 and S3, respectively, Zn element content is 63%(±0.3%), 55%(±0.7%) and 45% (±1.3%) for S1, S2 and S3, respectively.
2. Regarding the TEM analyses, the authors should write also in the experimental part of the manuscript (and not only in the cover letter) on how many particles the analyses was made, what kind of software they used and the error made in the measurements.
Per the Referee’s constructive comments, the manuscript has been modified as follows.
The morphology and particle sizes distribution of AuZnO composite nanoparticles(S1) were analyzed by TEM. As shown in the Fig. 1a, the composite nanoparticles are highly crystalline, uniform and nearly spherical. The histogram was obtained by counting a series of TEM images of the composite nanoparticles using Nano measurer software, as shown in the illustration, revealing that the composite nanoparticles have a narrow particle size distribution and consistent with the Gaussian distribution with an particle size of ~14.5(±0.9) nm. The TEM analysis give the particle size of ~13.8(±1.4) nm and ~15.3(±1.5) nm in diameter is for S2 and S3, respectively.
3. Regarding the TGA (and not TAG!) the authors write TGA curves but I see only one curve for S2. Where are the others? Are they similar? Please also report the first derivative of the curve in order to better individuate the weight losses.
Per the Referee’s constructive comments, the manuscript has been modified as follows.
Fig. S8 shows the TG-DTA curves of PEO-PPO-PEO terminated AuZnO(S2) composite nanoparticles. It can be seen from the TG curve in the figure that the weight loss of PEO-PPO-PEO capped AuZnO composite nanoparticles can be divided into three stages. From room temperature to 260 °C, weight loss of PEO-PPO-PEO-terminated AuZnO composite nanoparticles is 2%, mainly due to the removal of physically adsorbed water and chemisorption water [44]. The second stage weight loss is about 4%, corresponding to the volatilization and decomposition of PEO-PPO-PEO molecules adsorbed on the surface of the composite nanoparticles (between about 260 and 450 °C) [45]. In the third stage, there is no obvious weight loss. The differential thermal analysis (DTA) of samples was shown in Fig. S8. Three endothermal peaks 70 °C, 178 °C and 413°C correspond well to desorption of physically adsorbed water, removal of chemically adsorbed water and the volatilization and decomposition of PEO-PPO-PEO molecules adsorbed on the surface of the composite nanoparticles, respectively. On the DTA curve, a main exothermic effect was observed between 460 and 600 °C with a maximum at about 544 °C, indicating that the thermal events can be associated with the burnout of organic species involved in the precursor powders (organic mass remained from PEO-PPO-PEO), of the residual carbon or due to direct crystallization of AuZnO nanocrystalline from the amorphous component [46]. The TG-DTA curves of PEO-PPO-PEO-terminated AuZnO composite nanoparticles (S1, S2 and S3) are similar.
Fig. S8 TG-DTA analysis of PEO-PPO-PEO coated AuZnO(S2) composite nanoparticles from 25 to 600 °C.
4. In general, as evident reading any scientific book, figures must be embedded within the corresponding text. In other word the authors should write first a line or two of text and then the corresponding figures not the contrary
Per the Referee’s constructive comments, the manuscript has been modified and the revised manuscript is highlighted in blue.
Reviewer 4 Report
Accept in its present form.
Author Response
1. (x) English language and style are fine/minor spell check required
Per the Referee’s constructive comments, the manuscript has been modified and the revised manuscript is highlighted in blue.
Round 3
Reviewer 1 Report
Applied Sciences-MDPI
Manuscript ID: applsci-389879
Manuscript Type: article
Title: “Photocatalytic activity of monosized AuZnO composite nanoparticles”
Authors: Chenguang Ma, Xianhong Wang, Shixia Zhan, Xuemei Li, Xiao Liu, Yun Chai, Ruimin Xing, Hongling Liu
The authors have improved the experimental part with the required analyses however, curiously enough, while after the first revision the gold content for the three samples was reported to be 6, 8, 29% in this new version the gold content has changed into 7%(±0.3%), 13%(±1%) and 26%(±1%) for S1, S2 and S3. The authors should explain this difference (well outside the error range) between the two versions. Moreover they can not write 7%(±0.3%)!!! As described in any General or Analytical textbook they should write 7.0±0.3%.
Author Response
1. The authors have improved the experimental part with the required analyses however, curiously enough, while after the first revision the gold content for the three samples was reported to be 6, 8, 29% in this new version the gold content has changed into 7%(±0.3%), 13%(±1%) and 26%(±1%) for S1, S2 and S3. The authors should explain this difference (well outside the error range) between the two versions. Moreover they cannot write 7%(±0.3%)!!! As described in any General or Analytical textbook they should write 7.0±0.3%.
Per the Referee’s constructive comments, the manuscript has been modified as follows.
The elemental content of the AuZnO composite nanoparticles was tested by ICP analysis. The results show that Au element content is 7.0±0.3%, 13.1±1% and 26.1±1% for S1, S2 and S3, respectively, Zn element content is 63.0±0.3%, 55.1±0.7% and 45.0±1.3% for S1, S2 and S3, respectively.